# Structural and Biophysical Insights into SPINK1 Bound to Human Cationic Trypsin

**DOI:** 10.3390/ijms23073468

**Published:** 2022-03-23

**Authors:** Felix Nagel, Gottfried J. Palm, Norman Geist, Thomas C. R. McDonnell, Anne Susemihl, Britta Girbardt, Julia Mayerle, Markus M. Lerch, Michael Lammers, Mihaela Delcea

**Affiliations:** 1Biophysical Chemistry, Institute of Biochemistry, University of Greifswald, 17489 Greifswald, Germany; felix.nagel@uni-greifswald.de (F.N.); norman.geist@uni-greifswald.de (N.G.); anne.susemihl@uni-greifswald.de (A.S.); 2Synthetic and Structural Biochemistry, Institute of Biochemistry, University of Greifswald, 17489 Greifswald, Germany; palm@uni-greifswald.de (G.J.P.); bgirbard@uni-greifswald.de (B.G.); michael.lammers@uni-greifswald.de (M.L.); 3Biochemical Engineering Department, University College London, Bernard Katz, London WC1E 6BT, UK; thomas.mcdonnell.11@ucl.ac.uk; 4Department of Hematology and Oncology, Internal Medicine C, University of Greifswald, 17489 Greifswald, Germany; 5Department of Medicine II, University Hospital Munich, Ludwig-Maximillian University Munich, 81377 Munich, Germany; julia.mayerle@med.uni-muenchen.de; 6Department of Medicine A, University Medicine Greifswald, 17489 Greifswald, Germany; markus.lerch@med.uni-muenchen.de

**Keywords:** pancreas, pancreatitis, Kazal inhibitor, serine protease, N34S, protein–protein interaction, standard mechanism, catalytic triad, crystal structure, surface plasmon resonance (SPR), isothermal titration calorimetry (ITC), molecular dynamics simulations (MDS)

## Abstract

(1) The serine protease inhibitor Kazal type 1 (SPINK1) inhibits trypsin activity in zymogen granules of pancreatic acinar cells. Several mutations in the *SPINK1* gene are associated with acute recurrent pancreatitis (ARP) and chronic pancreatitis (CP). The most common variant is SPINK1 p.N34S. Although this mutation was identified two decades ago, the mechanism of action has remained elusive. (2) SPINK1 and human cationic trypsin (TRY1) were expressed in *E. coli*, and inhibitory activities were determined. Crystals of SPINK1–TRY1 complexes were grown by using the hanging-drop method, and phases were solved by molecular replacement. (3) Both SPINK1 variants show similar inhibitory behavior toward TRY1. The crystal structures are almost identical, with minor differences in the mutated loop. Both complexes show an unexpected rotamer conformation of the His63 residue in TRY1, which is a member of the catalytic triad. (4) The SPINK1 p.N34S mutation does not affect the inhibitory behavior or the overall structure of the protein. Therefore, the pathophysiological mechanism of action of the p.N34S variant cannot be explained mechanistically or structurally at the protein level. The observed histidine conformation is part of a mechanism for SPINK1 that can explain the exceptional proteolytic stability of this inhibitor.

## 1. Introduction

The Kazal-type serine protease inhibitor domain (protein family PF00050) presently consists of 9000 members and runs through all domains of life [1,2]. It is mostly indicative of protease inhibitors but has also been found in the extracellular regions of agrins, which are not known to inhibit proteases. The structure of a classical Kazal domain revolves around a central α-helix inserted between two β-strands and a third β-strand toward the C-terminus. In addition, three disulfide bridges are present in the majority of Kazal domains, stabilizing the overall structure and reactive loop [3]. The general inhibition mechanism of this protein family is competitive and temporary and is accompanied by reversible cleavage of the P1-P1′ scissile bond (Schechter and Berger nomenclature, Figure 1A) [4,5]. A prerequisite for the standard mechanism is for the inhibitor to be proteolytically stable enough to exert its function before it is ultimately degraded by its target protease [4,6,7,8]. This high proteolytic stability is achieved by low dissociation rates and re-ligation of cleaved scissile bonds, which can be catalyzed due to the absence of water at the interface, although the overall mechanism remains poorly understood [9,10,11,12,13]. 

In the present study, we investigated human serine protease inhibitor Kazal-type 1 (SPINK1) and its interaction with human cationic trypsin (TRY1). Two decades ago, the inhibitor first gained wider attention when a p.N34S amino acid substitution present in 2% of the population was found to be associated with acute recurrent pancreatitis (ARP) and chronic pancreatitis (CP) [14,15,16,17]. Prevalence for developing the disease is increased up to tenfold in carriers of the mutation, and the mechanism of action of this variant remains enigmatic to this day [18,19,20,21]. However, recent reviews of available genetic databases indicate a variant in linkage disequilibrium upstream of the *SPINK1* promoter to be causal of the disease [22]. Additionally, the *SPINK1* gene has been identified to be associated with many types of cancer [23,24,25], which is mostly attributed to potential epidermal growth factor receptor (EGFR) binding [26]. Furthermore, SPINK1 has been reported to inhibit granzyme A, preventing apoptosis in tumor cells [27]. 

For the past two decades, the structure of the SPINK1 complex with TRY1 has been under debate and was subjected to multiple molecular dynamics simulation studies [28,29], docking simulations [30], homology modeling [15,31] and spectroscopic studies, using circular dichroism [32,33]. Changes within the SPINK1 binding loop and peptide flipping due to the mutation were proposed. Unfortunately, these previous studies were inconclusive regarding differences between SPINK1 WT and the p.N34S mutant and/or did not provide the required resolution for a detailed molecular understanding of the interaction with TRY1.

Here we provide a structure–function analysis for the SPINK1–TRY1 interaction. Kinetic, thermodynamic and affinity binding data show a high affinity interaction with formation of very stable complexes for both SPINK1 WT and p.N34S mutant. We crystallized both SPINK1 variants in a complex with TRY1 and solved both structures at 2.9 and 2.1 Å resolution, respectively. Our structures differ from the aforementioned structural studies because the catalytic His63 adopts an outward facing conformation in both complexes. His63, thereby, faces away from Ser200 and Asp107 of the catalytic triad, rendering the protease inactive. The outward conformation may be further stabilized by a sulfate ion observed in close proximity to His63. We believe that the His63 conformation contributes significantly to the high proteolytic stability of the inhibitor. Hence, our results provide a structural platform that allows for the comparison of SPINK1 WT and the ARP and CP relevant p.N34S mutant bound to their natural target protease TRY1. Furthermore, they reveal a binding mechanism previously not known for this protein family and, thus, may have implications for other Kazal-inhibitors. 

## 2. Results

### 2.1. Inhibitory Activities of SPINK1 WT and p.N34S Are Similar

Despite it being clearly established that the p.N34S mutation in SPINK1 is associated with a ten-fold increase in prevalence for developing ARP or CP, the mechanism of action of this mutation has remained elusive [18]. To approach this problem, we expressed recombinant SPINK1 variants and human cationic trypsin (TRY1). Protein purity, homogeneity and complex formation were validated by analytical size exclusion chromatography, as well as SDS–PAGE (Figure 1B,C). 

Inhibitory constants (*K*_i_) were determined in equilibrium at 37 °C, using Morrison’s quadratic equation for tight binding inhibition utilizing previously determined *K*_m_ values (*K*_m_ = 1.2 mM; Figure 1D,E,J). Both SPINK1 variants exhibit *K*_i_ values in the sub-nanomolar range (*K*_i_ < 50 pM). Hence, the p.N34S amino acid substitution does not influence SPINK1 binding to TRY1 in equilibrium.

### 2.2. SPINK1 WT and p.N34S Show Similar Kinetics for Their Interaction with TRY1

Kinetic rate constants were determined by surface plasmon resonance (SPR). A CM5 sensor chip was prepared with covalently immobilized TRY1 p.S200A. The amino acid substitution of the catalytic serine residue was introduced to ensure surface stability and prevent autodigestion of the enzyme. The interaction with both SPINK1 variants was investigated by single-cycle kinetic experiments at 37 °C (Figure 1F,G,J). Kinetic rate constants were similar with a very high complex stability and association rates *k*_a_ of 2.5 × 10^5^ M^−1^ s^−1^. Both SPINK1 variants in complex with TRY1 p.S200A displayed a half-life *t*_1/2_ ~ 1 h. While a single point mutation can often cause similar equilibrium affinities but very different binding kinetics, this is not the case for the SPINK1–TRY1 interaction and the p.N34S amino acid substitution. 

### 2.3. SPINK1 WT and p.N34S Display Similar Thermodynamic Profiles for Their Interaction with TRY1

Isothermal titration calorimetry was carried out with TRY1 p.S200A placed in the sample cell and SPINK1 in the syringe. Dissociation constants (*K*_d_) were not determined, due to the high affinity of the interaction, resulting in c-values > 10,000. Without sufficient curvature, *K*_d_ cannot be assessed accurately, but valuable insights into the thermodynamics of the interaction can still be obtained. *K*_d_ values from SPR experiments were used for the thermodynamic analysis. Injections were carried out with appropriate spacing to ensure full equilibration before the subsequent injection (Figure 1H,I,J). Determined molar enthalpies for both SPINK1 variants were similar and revealed favorable binding enthalpies at approximately −10 kcal mol^−1^. The interaction is primarily enthalpically driven with little entropic contribution. Overall, both SPINK1 variants are functionally indistinguishable.

### 2.4. Structure of SPINK1 Variants Bound to TRY1

We solved the crystal structures of TRY1 p.S200A bound to SPINK1 WT and p.N34S at 2.9 and 2.1 Å resolution, respectively (Table 1 and Figure 2A). The final structure of TRY1 p.S200A contains amino acids 24–247 and amino acids 24–79 for the SPINK1 variants. TRY1 adopts the typical serine protease fold containing two β-barrels sandwiching the active site cleft of the protease. The S1 binding pocket determining the specificity of the protease contains the negatively charged Asp194, which, in turn, leads to the favorable binding of the positively charged Lys41 residue in SPINK1. The inhibitor assumes a typical Kazal domain structure with a reactive loop surrounding a central α-helix. The loop is stabilized by three disulfide bridges, and the domain contains a small portion of β-sheets. SPINK1 binds on top of the TRY1 p.S200A active site in a substrate-like manner, creating an interface of almost 1000 Å^2^ (Figure 2B and Appendix A). The structures of both SPINK1 variants are mostly similar, with small differences only present in the loop orientation of the p.N34S mutation site. However, due to high flexibility of this loop, side chain conformations are poorly defined, and changes are unlikely to translate into functional effects (Figure 2C). 

### 2.5. Interactions of SPINK1 Variants with TRY1 p.S200A

Similar to other known Laskowski-inhibitors, both SPINK1 variants bind to TRY1 p.S200A in a substrate-like manner. The specificity determining P1 residue Lys41 is oriented in a way that interacts with Asp194 in TRY1. The oxygen of the carbonyl carbon is stabilized by the oxyanion hole formed by the backbone nitrogen atoms of Ala200 and Gly198 (Figure 3A and Appendix A). In TRY1 WT, Ala200 would be replaced by a serine residue oriented in a way that allows for the cleavage between the P1-P1′ residues in both SPINK1 variants. Apart from the specific Lys41–Asp194 interaction, most other interactions involve the SPINK1 backbone. The loop orientation is also stabilized by intramolecular interactions within the SPINK1 inhibitor. Most prominently, Asn56 interacts with Thr40 and Ile42, which are both direct neighbors to Lys41 (Figure 3B,E and Appendix A). Asn56 is very conserved among Kazal domains and probably exerts similar functions in related Kazal inhibitors. Locking the P1′ and P2 residues into place, Asn56 most likely stabilizes the loop conformation even after cleavage of the P1-P1′ scissile bond, aiding the re-synthesis of the peptide bond. Furthermore, loop flexibility is limited by disulfide bridges, as well as the neighboring β-sheet toward the C-terminal end of the loop. Asn34 or Ser34 does not directly interact with TRY1. However, the neighboring Tyr33 residue engages in a cation–pi bond with Arg101 in TRY1. This interaction causes Tyr33 to be pulled away from the core interface, making room for a sulfate ion in the SPINK1 loop (Figure 3D and Appendix A). Coincidentally, a p.R101H mutation in rat anionic trypsin was described that creates a metal binding site, causing His63 to assume an energetically more favorable *trans* conformation [34].

### 2.6. Conformational Change in the TRY1 Catalytic Triad

The structures of the active site regions of the enzyme–inhibitor complex resemble that of a typical Kazal inhibitor–serine protease complex. The scissile peptide group displays a planar geometry with the carbonyl-oxygen pointing toward the oxyanion hole. In TRY1 WT, the carbonyl carbon would be placed ideally for the generation of the acyl-intermediate with Ser200. In the present structure, Ser200 is replaced by an alanine to avoid proteolytic cleavage of the inhibitor or the protease due to residual protease activity observed even in the inhibitor bound complex. Asp107 is positioned in a way that would stabilize the His63 residue in the wild-type protease. Unexpectedly, His63 exhibits an uncommon rotamer conformation, pointing outward and away from the catalytic triad (Figure 3D and Appendix A). Instead, it points toward the sulfate ion bound within the SPINK1 loop. The negative charge of the sulfate ion may compensate for the missing charge of Asp107 in the observed histidine conformation, while a water molecule is located near Asp107. The *trans* conformation is further stabilized by Tyr99 in TRY1 and Thr40 in SPINK1. With this uncommon histidine conformation, TRY1 would not be able to catalyze the cleavage of the P1-P1′ scissile bond, as His63 could not function as a proton acceptor for Ser200, thereby not creating the strongly nucleophilic alkoxide ion. While the p.S200A mutation might destabilize the His63 conformation, many serine protease crystal structures have been solved that contain the same p.S200A mutation, without affecting the histidine conformation (List S1) [9,36,37]. To analyze the impact of SPINK1 binding on the catalytic triad, we employed molecular dynamics simulations by using TRY1 WT and SPINK1 WT complexes, with His63 starting in either *gauche*^+^ or *trans* (Chi_1_ = 60° or 180°) conformation. SPINK1 binding sterically blocks the His63 rotamer conformation to flip from one state to the other most of the time (Figure 4A). During the simulation, we observed one *gauche*^+^ to *trans* and *trans* to *gauche*^+^ transition, which was preceded by a rotamer change of Thr40 in SPINK1 (Figure 4B,C and Appendix A). In the *gauche*^+^ conformation, Ser200 and Asp107 are separated by 7.8 Å, while the distance is reduced to 7.3 Å in the *trans* conformation, potentially blocking easy re-entry of His63 into a productive catalytic triad arrangement (Appendix A). Additionally, no relevant crystal contacts are in close proximity to the flipped histidine and the adjacent SPINK1 and TRY1 loops. Therefore, our data provide support for a model in which trypsin inhibition by SPINK1 is in concert with conformational changes within the trypsin active site. 

## 3. Discussion

We demonstrated that SPINK1 WT and the p.N34S mutant display similar inhibition constants, binding kinetics and thermodynamic properties upon binding human cationic trypsin (TRY1). This behavior is already well-known within the pancreatic community and has been shown in numerous studies [32,38,39]. Our determined *K*_d_ values and association rate constants are higher compared to values reported by Szabó et al. [31] but lower than those reported by Király et al. [38]. Still, all studies show sub-nanomolar affinities and are therefore essentially in agreement with one another. Possibly due to their similar properties, most carriers of the p.N34S mutation do not develop ARP or CP throughout their entire life [40], despite the prevalence for developing idiopathic chronic pancreatitis (ICP) being increased tenfold. Due to inhibitory properties not being able to explain the pathogenicity of the p.N34S variant, the call for a detailed atomistic model emerged already two decades ago [15]. While several homology models and molecular dynamic simulation studies aimed to meet the demand for such a detailed structure, they are all accompanied by uncertainties, whether a simulated model can accurately reflect reality [41]. Our crystal structures of SPINK1 WT and p.N34S in complex with TRY1 p.S200A provide experimentally determined, high-resolution structures and represent the first structures, where a human Kazal-inhibitor is bound to its natural target protease.

By comparing both structures, we show a slight tilt of the p.N34S variant’s α-helix. However, apart from the Arg101–Tyr33 interaction, the loop which contains the p.N34S mutation does not engage in any interactions with TRY1, explaining the high flexibility and limited influence on the interaction of this loop (Figure 2C). While we show minor differences between both structures, they do not translate into different binding constants or other functional differences. Nevertheless, they might contribute to SPINK1 p.N34S being a risk-factor for ARP or CP by other mechanisms. Recently, decreased binding affinity of SPINK1 toward TRY1 was demonstrated after sulfation of Tyr154 in TRY1 [31]. Due to sulfation of the tyrosine, a steric clash with Tyr43 and Pro55 in SPINK1 was proposed, and our crystal structures support this hypothesis (Appendix A).

His63 in TRY1 adopts a *trans* conformation rarely found in serine proteases instead of the *gauche^+^* conformation [42]. In this conformation, His63 is facing out and away from the catalytic triad. Therefore, the protease’s activity is reduced, missing the necessary proton acceptor for Ser200. The remaining members of the catalytic triad remain in their typical conformation. In this study, Ser200 is replaced by an alanine to render the protease inactive for crystallization. The introduced mutation and the crystallization process itself can introduce artefacts that might cause His63 to flip into the *trans* conformation. However, in combination with molecular dynamics simulations, we demonstrate that SPINK1 binding sterically hinders the *trans* to *gauche*^+^ transition and can affect the His63 conformation. The rotamer change is preceded by a conformational change of Thr40 in SPINK1, and a stabilizing water molecule between Ser200 and Asp107 enters the space usually occupied by His63. The sulfate ion in the crystal structure probably further stabilizes the *trans* conformation and could shift the equilibrium toward the *trans* conformation. Phosphorylation of Ser34 might accomplish the same effect; however, there is currently no indication for post translational modifications in SPINK1. Conversely, a related structure of porcine SPINK1 bound to bovine trypsin with sulfate present displays the *gauche*^+^ conformation, and other structures containing the p.S200A mutation also show the *gauche*^+^ conformation [43].

His63 is one of the most important residues in serine protease catalysis [42]. It functions as a proton donor and acceptor throughout the mechanism and allows for the formation of the strongly nucleophilic alkoxide ion. The Ser-His dyad is very conserved among serine proteases and evolved independently numerous times [44,45]. Even though His63 is indispensable for efficient catalysis, substitution by an alanine does not render the protease completely inactive, and, rather, *k*_cat_ values decrease by three to four orders of magnitude [46,47]. Thus, we assume that the observed conformation of His63 does not inactivate the protease completely, but lowers the rate of SPINK1 proteolysis. Another contributing factor for the proteolytic stability of SPINK1 is Asn56, which bridges the P1′ and P2 residues by hydrogen-bonding. Thereby, the binding loop of SPINK1 likely remains rigid even after cleavage of the P1-P1′ scissile bond.

## 4. Conclusions

Our structure–function analysis provides insights into the molecular mechanism, as well as structure, of the SPINK1–TRY1 complex and the p.N34S variant. While we were unable to elucidate the underlying cause of the p.N34S pathogenicity in ARP and CP, we still found minor differences in their complex structures. Previously, structural differences between both SPINK1 variants were hypothesized to be disease causing or modifying [15,28]. However, our crystal structures prove that both SPINK1 variants are mostly similar, and the minor differences we detected are unlikely to translate into functional or clinical effects.

We are thereby further solidifying the current view of a variant upstream of *SPINK1* promoter being the disease-causing mutation, rather than the p.N34S mutation itself [22]. Moreover, our data may prove useful in the field of cancer research, as the roles of SPINK1 in EGFR binding and granzyme A inhibition are still poorly understood. 

In addition, complex formation of both SPINK1 variants is accompanied by flipping of the His63 residue within the catalytic triad. In non-complexed TRY1, His63 adopts a *gauche^+^* conformation and adopts a *trans* conformation upon complex formation with SPINK1. This mechanism is atypical for Kazal-inhibitors or other proteinaceous inhibitors in general. We speculate that, by this mechanism, SPINK1 elegantly evades proteolysis by TRY1 and thereby displays an increased half-life. Further studies with similar Kazal-inhibitors are planned and are aimed at elucidating whether the observed conformational change in TRY1 upon SPINK1 binding is a mechanism shared among other inhibitors within this family. Such studies will aid in rationalizing the exceptional proteolytic stability of these inhibitors, while also providing a new platform for developing rationally designed proteinaceous protease inhibitors. 

## 5. Materials and Methods

If not otherwise indicated, chemicals were purchased from Sigma (Sigma-Aldrich, Taufkirchen, Germany). 

### 5.1. Protein Expression and Purification

The cDNA coding for amino acids 24-79 of SPINK1 or amino acids 16-247 of TRY1 was cloned into a pET47b expression vector (Novagen, Darmstadt, Germany) in frame with an N-terminal hexahistidine tag (His_6_-tag) and an HRV3C cleavage site. The N-terminal truncations were introduced to remove the signal peptide sequences for expression in *E. coli*. Mutations were generated by site directed mutagenesis, using QuikChange XL (Agilent Technologies, Santa Clara, CA, USA). 

SPINK1 variants were overexpressed in SHuffle T7 Express *E. coli* (New England Biolabs, Frankfurt am Main, Germany) grown in terrific broth medium supplemented with 100 µg mL^−1^ kanamycin at 30 °C. After 16 h of induction at an OD_600_ = 2 with 1 mM isopropyl-β-d-thiogalactoside (IPTG) at 16 °C, cells were harvested by centrifugation and resuspended in lysis buffer containing 20 mM Hepes pH 8, 150 mM NaCl and 20 mM imidazole before lysis by sonication. Clarified lysates were loaded onto a HisTrap HP (Cytiva) affinity column. The column was washed with lysis buffer containing 50 mM imidazole and eluted with lysis buffer containing 500 mM imidazole. The purified SPINK1 was subsequently dialyzed into 20 mM Hepes pH 8, 150 mM NaCl and digested with HRV3C protease (Merck, Darmstadt, Germany). The His_6_-tag was removed by using the same HisTrap column.

TRY1 variants were overexpressed in NiCo21 (DE3) *E. coli* (New England Biolabs, Frankfurt am Main, Germany) grown in terrific broth medium supplemented with 100 µg mL^−1^ kanamycin at 37 °C. After 16 h of induction at an OD_600_ = 2 with 1 mM IPTG at 25 °C, cells were harvested by centrifugation and resuspended in 0.1 M Tris pH 8 and 5 mM EDTA (Tris-EDTA). Inclusion bodies were prepared as described elsewhere [48]. In brief, inclusion bodies were washed by using Tris-EDTA three times before solubilization in 6 M guanidine hydrochloride (Gdn-HCl), 0.1 M Tris pH 8, 2 mM EDTA and 30 mM dithiothreitol (DTT). After 30 min of incubation at 37 °C and subsequent centrifugation, solubilized inclusion bodies were added to 0.9 M Gdn-HCl, 0.1 M Tris pH 8, 2 mM EDTA, 1 mM L-cystine, 1 mM L-cysteine at a speed of 20 µL min^−1^, using a syringe pump at 4 °C. After incubation overnight, refolded TRY1 was loaded onto a HisTrap excel (Cytiva, Freiburg, Germany) column. The column was extensively washed with lysis buffer and eluted by using 500 mM imidazole. Purified TRY1 was dialyzed against 10 mM Hepes pH 8, 75 mM NaCl and 2 mM CaCl_2_ before digestion by enterokinase (Genescript, Leiden, The Netherlands). His_6_-tag and propeptides were removed by size-exclusion chromatography, using a Superdex75 Increase 10/300 GL column (Cytiva, Freiburg, Germany) with 20 mM Hepes pH 8 and 150 mM NaCl. 

### 5.2. Enzyme Inhibition Studies

Inhibition constants (*K*_i_) of SPINK1 WT and N34S mutant were determined by monitoring the conversion of *N*_α_-Benzoyl-L-arginine 4-nitroanilide hydrochloride (L-BAPA, Bachem, Bubendorf, Switzerland) in the presence and absence of varying inhibitor concentrations (15.625–2000 pM) at 405 nm, using a Cytation 5 microplate reader (BioTek, VT, USA). The Michaelis–Menten kinetics of L-BAPA with human cationic trypsin was determined by following the reaction with varying substrate concentrations from 0.02 to 9.5 mM (Equation (1)).
(1)$$v=\frac{v_{max}\;\left[S\right]}{K_{m}+\left[S\right]}$$

Initial conversion rates were measured by using 1 mM L-BAPA, 100 pM TRY1 and varying SPINK1 concentrations in 20 mM Hepes pH 7.4, 150 mM NaCl, 2 mM CaCl_2_ and 0.05% Tween20. All assays were carried out at 37 °C, and initial rates were determined after a 5 h lag phase to ensure sufficient equilibration. Data were fitted by using Morrison’s quadratic equation for tight binding inhibitors (Equation (2)).
(2)$$v=v_{0}\left(1-\frac{\left(\left[E\right]+\left[I\right]+K_{i}(1+\left(\frac{\left[S\right]}{K_{m}}\right)\right)-\sqrt{{\left(\left[E\right]+\left[I\right]+K_{i}(1+\left(\frac{\left[S\right]}{K_{m}}\right)\right)}^{2}-4\left[E\right]\cdot \left[I\right]}}{2\left[E\right]}\right)$$

Reported *K*_i_ values were determined from fitting the averages of three independent experiments, and standard errors are shown.

### 5.3. Surface Plasmon Resonance

Surface plasmon resonance (SPR) kinetic analyses were carried out by using a BIAcore T200 instrument (Cytiva, Freiburg, Germany) at 37 °C and a flow rate of 50 µL min^−1^, and data were recorded at 10 Hz. Standard amine coupling chemistry ((3-dimethylaminopropyl)-3-ethylcarbodiimide(EDC)/N-hydroxysuccinimide(NHS)) was used according to the manufacturer’s instructions to covalently immobilize TRY1 p.S200A to a CM5 sensor chip (Cytiva, Freiburg, Germany). 

Concentration series of SPINK1 variants were prepared by two-fold dilutions in running buffer (20 mM Hepes pH 7.4, 150 mM NaCl, 2 mM CaCl_2_, 0.05% Tween20), ranging from 3.125 to 50 nM. Sensorgrams were recorded as single-cycle experiments and were double referenced. After each single-cycle experiment, the surface was regenerated by using 10 mM glycine pH 1.4. For kinetic analyses, sensorgrams were fitted to a 1:1 Langmuir model. Reported kinetic rate constants and dissociation constants represent average and standard deviation of at least three independent experiments. 

### 5.4. Isothermal Titration Calorimetry

Isothermal titration calorimetry (ITC) experiments were carried out by using a MicroCal PEAQ-ITC (Malvern Panalytical, Herrenberg, Germany) instrument at 37 °C and a reference power of 3 µcal s^−1^ in 20 mM Hepes at pH 7.4, 150 mM NaCl and 2 mM CaCl_2_. Buffers were degassed to ensure signal stability. Delay between injections was set to 240 s. The first injection was rejected from analysis. Subsequently, 18 injections of 3 µL each were carried out. TRY1 p.S200A was placed in the sample cell at 5–10 µM, and SPINK1 was in the syringe at 50–100 µM. Molar binding enthalpies were determined by peak integration, and the heat of dilution was determined from the titration steps at the end of the experiment and subsequently subtracted. The isotherms were fitted to a one-site binding model. Determined parameters are averages and standard deviations of at least three independent experiments. 

### 5.5. Crystallization and Data Collection

SPINK1–TRY1 p.S200A complexes were concentrated to 20 mg mL^−1^ and crystallized by hanging drop vapor diffusion at 20 °C, using a reservoir solution of 15% PEG 4000 and 0.3 M (NH_4_)_2_SO_4_. Crystals between 0.2 and 0.3 mm were soaked briefly in reservoir solution supplemented with 8% PEG 400 as cryoprotectant before cryocooling in liquid nitrogen. Diffraction data were collected at BESSY beamline 14.1 (SPINK1 WT) or 14.2 (SPINK1 N34S). Data collection parameters are given in Table 1.

### 5.6. Structure Determination

Data reduction was carried out in XDS [49], and phases were solved by molecular replacement in Phaser [50], using human cationic trypsin [51] and a mutated SPINK1 [52] variant as search models (pdb: 1TRN and 1CGI). Refinement was performed by using Refmac5 for the TRY1 p.S200A–SPINK1 N34S complex [53] and in Phenix for the TRY1 p.S200A–SPINK1 WT complex [54]. The higher resolution TRY1 p.S200A–SPINK1 N34S was used as a reference model for weighting restraints during refinement of the SPINK1 WT complex. The model was built in Coot [55]. The crystals are isomorphous and contain two heterodimers per unit cell. NCS restraints were not applied. TLS refinement was used for both structures. Refinement parameters are given in Table 1.

### 5.7. Molecular Dynamics Simulations

Molecular dynamics simulations of the WT TRY1–SPINK1 complex and His63 starting in either *trans* or *gauche*^+^ conformations were conducted with NAMD 2.14 [56]. The protein model was built from our WT crystal structure, without the sulfate ion. Force field parameters (CHARMM36) and mutations were added with CHARMM-GUI [57,58]. VMD 1.9.4 was utilized for hydrogen mass repartitioning on the protein and for adding water solvation (TIP3P) and ions to the cubic cell with side length 7 nm [59,60]. Physiological sodium chloride concentration was added at 0.15 mol/L, and the system was neutralized accordingly. The simulation time step was set to 4 fs. Pressures and temperatures were adjusted by a Langevin piston barostat at 100 fs period, 200 fs decay time and Langevin thermostat at 1 ps^−1^ damping time, respectively. The SETTLE algorithm constrained covalent bonds involving hydrogen atoms. Long-range electrostatic interactions were described by the smooth particle mesh Ewald method (sPME), while short-range interactions cutoffs were set to 1 nm with a switching function of 0.1 nm. The system was first subjected to standard energy minimization for 50 k steps and then slowly heated from 100 to 300 K, in steps of 50 K, for 1 ns each. Afterward, the final production runs proceeded at 310 K in an NPT ensemble for 5847 ns and 5048 ns, for *trans* or *gauche*^+^ conditions, respectively. Samples were collected every 20 ps. Rotational state densities for His63 were extracted with VMD.

## Figures and Tables

**Figure 1 ijms-23-03468-f001:**
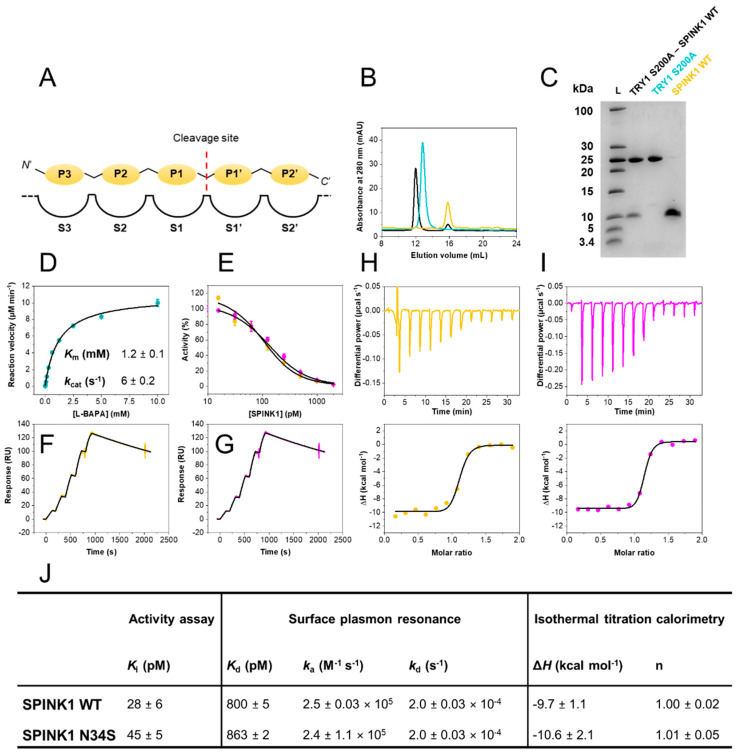
Biophysical characterization of the SPINK1–TRY1 interaction. (**A**) Schechter and Berger nomenclature. (**B**) Size-exclusion chromatograms of purified SPINK1 WT (yellow), TRY1 p.S200A (cyan) and complex (black). (**C**) 16% tricine SDS–PAGE of purified TRY1 p.S200A–SPINK1 WT complex, TRY1 p.S200A and SPINK1 WT. (**D**) Michaelis–Menten kinetics of L-BAPA substrate with TRY1. Error bars represent standard deviations of three independent experiments. *K*_m_ and *k*_cat_ values represent fitted values and their standard error. (**E**) Trypsin activity assay at varying SPINK1 concentrations fitted with Morrison’s quadratic equation. Error bars represent standard deviations of three independent experiments. (**F**) Surface plasmon resonance single-cycle kinetic of the TRY1 p.S200A—SPINK1 WT or (**G**) p.N34S interaction fitted with a 1:1 Langmuir interaction model. (**H**) Isothermal titration calorimetry of the TRY1 p.S200A–SPINK1 WT or (**I**) p.N34S interaction fitted with a 1:1 binding site model. (**J**) Summary of equilibrium, kinetic and thermodynamic data. *K*_i_ values represent fitted values ± standard errors, while SPR and ITC data are reported as mean ± SD of at least three independent experiments.

**Figure 2 ijms-23-03468-f002:**
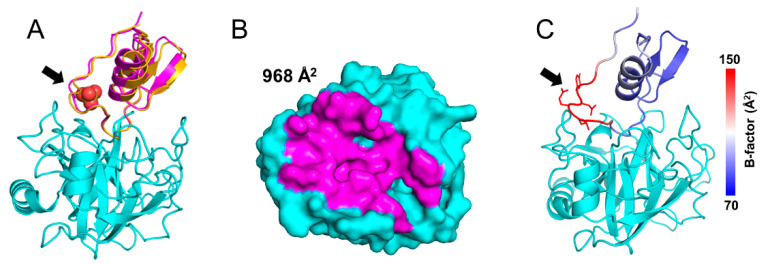
Structure and superposition of TRY1 p.S200A–SPINK1 complexes. The black arrow indicates the position of the p.N34S mutation site. (**A**) Structures and superposition of TRY1 p.S200A (cyan) in complex with SPINK1 WT (yellow) or p.N34S (pink). Sulfate ions are shown as spheres and are colored according to their atom type. (**B**) Binding interface of SPINK1 p.N34S. Due to their similarity, the binding interface of SPINK1 WT was omitted but can be seen in Appendix A. (**C**) SPINK1 WT in complex with TRY1 colored by B-factors.

**Figure 3 ijms-23-03468-f003:**
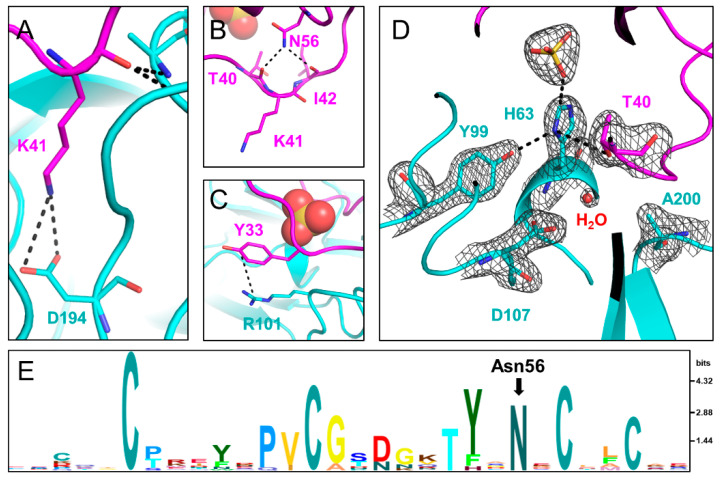
Interactions between TRY1 p.S200A (cyan) and SPINK1 p.N34S (pink). (**A**) Lys41 in SPINK1 interacts with the specificity determining Asp194 in TRY1. (**B**) Asn56 in SPINK1 stabilizes the binding loop by hydrogen bonding with Thr40 and Ile42. (**C**) Tyr33 in SPINK1 forms a cation—pi bond with Arg101 in TRY1 and is hence pulled outward. (**D**) Catalytic triad of the TRY1 p.S200A–SPINK1 complex (cyan). The 2Fo–Fc density map is shown at 1.6 Å around the residues of the catalytic triad and is contoured at 1.5 σ. In the complex structure His63 is rotated toward the sulfate ion and out of the productive catalytic triad arrangement. (**E**) Sequence conservation of the Kazal 1 family displayed by an HMM logo generated in Skylign [35]. Amino acid letter height is calculated based on the information content above background expressed in bits.

**Figure 4 ijms-23-03468-f004:**
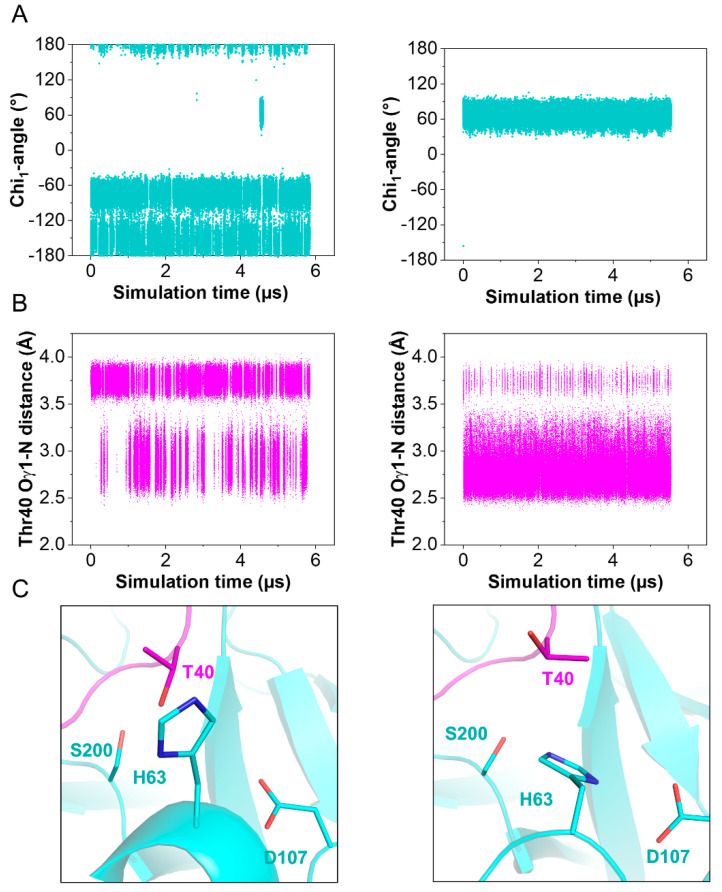
Observed His63 conformations during molecular dynamics simulations. Simulations started with His63 in either *trans* (Chi_1_ = 180°) or *gauche*^+^ (Chi_1_ = −60°) conformation (**left** and **right** panel, respectively) (**A**) Chi_1_—angles of His63. (**B**) Oγ1—N distance of Thr40 in SPINK1. (**C**) Representative side chain conformations of Thr40 and His63.

**Table 1 ijms-23-03468-t001:** Data collection and refinement of SPINK1–TRY1 p.S200A complexes.

	SPINK1 WT–TRY1 p.S200A	SPINK1 p.N34S–TRY1 p.S200A
**Data collection**		
Beamline	14.1 at BESSY	14.2 at BESSY
Wavelength (Å)	0.9184	0.9184
Unit-cell parameters (Å) *a*, *c* in space group P3_1_21	77.63, 187.35	76.55, 189.72
Resolution (Å)	50–2.90 (3.08–2.90)	50–2.10 (2.22–2.10)
No. of unique reflections (Friedel pairs merged)	15,112 (2371)	38,593 (6081)
Redundancy	19.1 (19.4)	19.7 (19.4)
Completeness (%)	99.8 (99.0)	99.8 (98.7)
R_merge_	0.166 (2.651)	0.174 (3.172)
cc_1/2_	0.999 (0.550)	0.999 (0.411)
<I/σ(I)>	16.3 (1.2)	14.8 (1.0)
Wilson *B*-factor (Å^2^)	85.6	50.1
**Refinement**		
Resolution range (Å)	50–2.90 (2.99–2.90)	50–2.10 (2.15–2.10)
Completeness (%)	99.7 (98.0)	99.8 (97.1)
No. of reflections, working set	13589 (1191)	36688 (2600)
No. of reflections, test set	1509 (133)	1904 (134)
Final *R*_work_	0.222 (0.365)	0.197 (0.334)
Final *R*_free_	0.243 (0.374)	0.233 (0.421)
No. of non-H atoms		
Protein	4219	4269
Solvent	16	236
R.m.s. deviations		
Bond lengths (Å)	0.011	0.008
Angles (°)	1.619	1.505
Average *B* factors (Å^2^)		
Protein	100.01	36.73
Solvent	138.68	51.09
Molprobity analysis		
Ramachandran most favored (%)	96.17	96.75
Ramachandran outliers (%)	0.0	0.0
Overall score	1.90	1.80
Clash score	13.48	3.80
PDB entry	7QE8	7QE9

## Data Availability

The structures were deposited in the protein data bank under the following ID codes: SPINK1 WT–TRY1 S200A, 7QE8; SPINK1 N34S–TRY1 S200A, 7QE9.

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
