# Peer review of "Structural and Biophysical Insights into SPINK1 Bound to Human Cationic Trypsin"

_ijms, 2022, doi:10.3390/ijms23073468_

Round 1
Reviewer 1 Report
The manuscript entitled “Structural and Biophysical Insights into SPINK1 Bound to Human Cationic Trypsin” is an interesting work. A lot of experimental work has been done but few issues need to be addressed before it can be accepted for publication.
- Two decades ago, the inhibitor first gained increased attention. Both the words gained and increased shouldn’t be used simultaneously.
- Dissociation constants (Kd) were not determined due to the high affinity of the interaction. Then, why the authors did perform ITC. ITC is amongst the most sensitive technique used to study binding energetics and thermodynamic parameters. It remains uncertain so as to what was the need to perform this experiment?
- Did the authors perform degasing to ensure signal stability during the ITC experiment?
- The ITC isotherms were plotted using how many model sites?
- Michaelis-Menten kinetics of L-BAPA substrate with TRY1. Check the flaw and correct it.
- Inhibitory constants (Ki) were determined in equilibrium at 37 °C using Morrison’s quadratic equation for tight binding inhibition utilizing previously determined Km values. The equation should be mentioned here. Also all the thermodynamic symbols need to be italicized throughout the manuscript.
- Figure 4 legend. The left pane started with His63 , while the right pane. Correct the legend.
- Our determined Kd – values and association rate constants are higher compared to [31] but lower compared to [38]. This is not a way to write. Rewrite the same.
- There is a lot of work done in the manuscript. This study was carried out to elucidate the mechanism of action but that has not been achieved. The authors should provide clearly in the conclusion section as how this study is novel and how will it help in elucidation of underlying cause of the p.N34S pathogenicity in ARP and CP.
- Molar binding enthalpies were determined by peak integration and heat of dilutions were determined from the titration steps at the end of the experiment and subsequently subtracted. The author should cite relevant literature for the same related to ITC experiment.
https://doi.org/10.3390/ijms222010986
- In the introduction section I feel a little paragraph about the proteases and the importance of proteases in context of various diseases can be added. Adding few of these literatures will aid in this.
https://doi.org/10.1016/j.ijbiomac.2017.04.071
Author Response
We thank the Reviewer for the comments.

Reviewer 2 Report
The authors describe a novel complex between human trypsin TRY1 and its inhibitor SPINK1 (serine protease inhibitor Kazal type 1) and SPINK1 pathogenic variant p.N34S. The mutant SPINK p.N34S is of clinical importance because is linked with increased risk of acute or chronic pancreatitis. The paper settles long suspected and debated assumption that pathogenicity of SPINK1 p.N34S variant is not due to its differential binding, inhibition or different functional properties compared to wt variant. Both wt and p.N34S SPINK1 variants bind to trypsin in essentially the same manner, according to the solved crystal structures. The structural investigations were complemented and corroborated by extensive kinetic (enzyme activity and inhibition assay), biophysical (kinetics and thermodynamics of inhibitor binding to trypsin) and computational (molecular dynamics simulations) characterisation.
The presented research is sound, comprehensive, of scientific and clinical relevance. As a reviewer, I have no concerns or objections whatsoever and I recommend the manuscript to be published in present form.
Author Response
We thank the Reviewer for the comments.

Reviewer 3 Report
Dear Authors,
Yours article entitled "Structural and Biophysical Insights into SPINK1 Bound to Human Cationic Trypsin" is well executed and written. The results and conclusions have been obtained using the adequate contemporary biochemical and biophysical methods and seem to be useful to understand the molecular mechanism of some pancreas diseases. I have not any essential remarks and recommend to accept the article in the present form.
Author Response
We thank the Reviewer for the comments.
